# Contributions of Organic and Mineral Matter to Vertical Accretion in Tidal Wetlands across a Chesapeake Bay Subestuary

Jenny R. Allen [1,*], Jeffrey C. Cornwell [2] and Andrew H. Baldwin [1]

1   Department of Environmental Science and Technology, University of Maryland, College Park, MD 20742, USA; baldwin@umd.edu
2   Horn Point Laboratory, University of Maryland Center for Environmental Science, Cambridge, MD 21613, USA; cornwell@umces.edu
*   Correspondence: jennyrallen@gmail.com

**Abstract:** Persistence of tidal wetlands under conditions of sea level rise depends on vertical accretion of organic and inorganic matter, which vary in their relative abundance across estuarine gradients. We examined the relative contribution of organic and inorganic matter to vertical soil accretion using lead-210 ($^{210}$Pb) dating of soil cores collected in tidal wetlands spanning a tidal freshwater to brackish gradient across a Chesapeake Bay subestuary. Only 8 out of the 15 subsites had accretion rates higher than relative sea level rise for the area, with the lowest rates of accretion found in oligohaline marshes in the middle of the subestuary. The mass accumulation of organic and inorganic matter was similar and related ($R^2 = 0.37$). However, owing to its lower density, organic matter contributed 1.5–3 times more toward vertical accretion than inorganic matter. Furthermore, water/porespace associated with organic matter accounted for 82%–94% of the total vertical accretion. These findings demonstrate the key role of organic matter in the persistence of coastal wetlands with low mineral sediment supply, particularly mid-estuary oligohaline marshes.

**Keywords:** coastal wetlands; accretion; organic matter; inorganic matter; sea level rise

## 1. Introduction

Tidal wetlands are under an increasing threat from rising sea levels. On average, global mean sea level rose approximately 1.2 mm yr$^{-1}$ between 1901 and 1990 and 3.1 mm yr$^{-1}$ between 1993 to 2017 [1–4]. Along the east coast of the US, a 1000 km stretch of coastline north of Cape Hatteras has been identified as a "hotspot" for accelerated sea level rise [5]. This includes the Chesapeake Bay region, where relative sea level rise (RSLR) rates ranged from 3.24–5.11 mm yr$^{-1}$ between 1969 and 2014 and have been accelerating by 0.08–0.22 mm yr$^{-1}$ [6]. Regional land subsidence plays a major role, with approximately 53% of RSLR due to local subsidence within the Chesapeake Bay [7].

The ability of tidal wetlands to maintain surface elevation under accelerated sea level rise is critical for their persistence and depends on many factors including accretion of mineral and organic matter, rates of decomposition, plant community structure, and productivity [8–11]. Marsh accretion is a balance between deposition and erosion on the marsh surface and belowground plant production and decomposition of organic materials [8,12,13]. These processes may vary across the range of tidal wetlands types and within individual wetlands as the sediment load changes and as wetland plant communities shape these dynamics [14–16].

The overall response of tidal wetlands to RSLR is dependent on the deposition of inorganic and organic matter [11,12,17], and the supply of fluvial or estuarine mineral sediment dictates the balance of organic and inorganic contributions to accretion. For example, in regions where sediment supply is low, such as in Louisiana and the Northeast

U.S., organic matter accumulation drives vertical accretion [18–20]. In the Southeast U.S., where sediment supplies are greater, the marshes are more mineral-rich [21]. However, the maximum rate of organic matter accretion is limited by plant productivity, so although organic matter accumulation drives accretion in some marshes [15,17,20], organic matter accumulation alone may not be enough to keep pace with sea level in some tidal wetland systems [13,21,22].

Accretion dynamics vary depending on the position of tidal marshes along the estuary. Previous studies have found that accretion rates often decrease with increasing salinity in estuarine systems along the Atlantic coast of the U.S. [23–25], indicating that rivers are a major source of suspended sediment supplied to these systems. However, mid-Atlantic U.S. estuaries are microtidal (mean tidal range < 2 m) and characterized by weak tidal current velocities with little capacity to transport suspended sediment [26]. Within smaller tributary estuaries, river dynamics may overwhelm the weaker tidal forces and develop a pronounced tidal asymmetry that creates a dominant ebb tide and net sediment export [27]. As a result, accretion rates are typically highest in the upper reaches of the estuary and decrease downstream as suspended sediment decreases.

The overall goal of this study was to evaluate accretion rates in tidal wetlands across a Chesapeake Bay subestuary salinity gradient using $^{210}$Pb dating of soil cores, a method commonly used in wetlands to determine accretion rates over an approximately 100 year timeframe [15,18,22]. Specifically, the objectives of this study were to examine accretion dynamics between marshes across the estuarine gradient and to determine the relative contribution of organic and inorganic matter to accretion in these marshes. We hypothesized that rates of accretion will be highest in the tidal freshwater marshes and decrease down the estuary due to less fluvial sediment input. We also expected that the relative contribution of organic matter to accretion would increase as mineral sediment input declined proceeding downstream across the estuary.

## 2. Materials and Methods

The Nanticoke River, a major tributary estuary of the Chesapeake Bay, drains over 200,000 hectares of Maryland and Delaware's coastal plain in the central Delmarva Peninsula [28]. The main stem of the river flows southwest from southern Delaware through Maryland into Tangier Sound. The river widens proceeding downstream, and is bordered by expansive tidal wetlands, including tidal forested wetlands and estuarine-meander marshes in the upper estuary and submerged-upland marshes in the lower estuary [29]. The estuary is ebb-dominated and microtidal, with a mean tidal range of 0.7 m (NOAA water level station 8571773, Vienna, MD, USA). Salinity ranges from approximately 15 ppt at the mouth to less than 0.5 ppt in the tidal freshwater zone [30]. The watershed is dominated by agriculture (39.2%) and forested areas (40.9%) [28,31]. The Nature Conservancy designated the Nanticoke River watershed as a bioreserve and a "Last Great Place" in 1991. Within the watershed, there are approximately 200 plant species and 70 animal species listed as rare, threatened or endangered, and 20 plant and 5 animal species that are globally rare [32].

Five sites were selected along the salinity gradient of the Nanticoke River and three subsites were established within interior marsh sections of each site (Figure 1) [30]. Sites 1 and 2 were located downstream in mesohaline marshes, sites 3 and 4 were located within oligohaline marshes, and site 5 was located upstream in tidal freshwater marshes. Additionally, sites 1–3 were located in large marshes with extensive tidal creeks, while sites 4 and 5 were located in smaller marshes closer to the main river channel. Plant species composition varies from brackish to freshwater species as salinity decreases proceeding upstream across the estuary (Table 1).

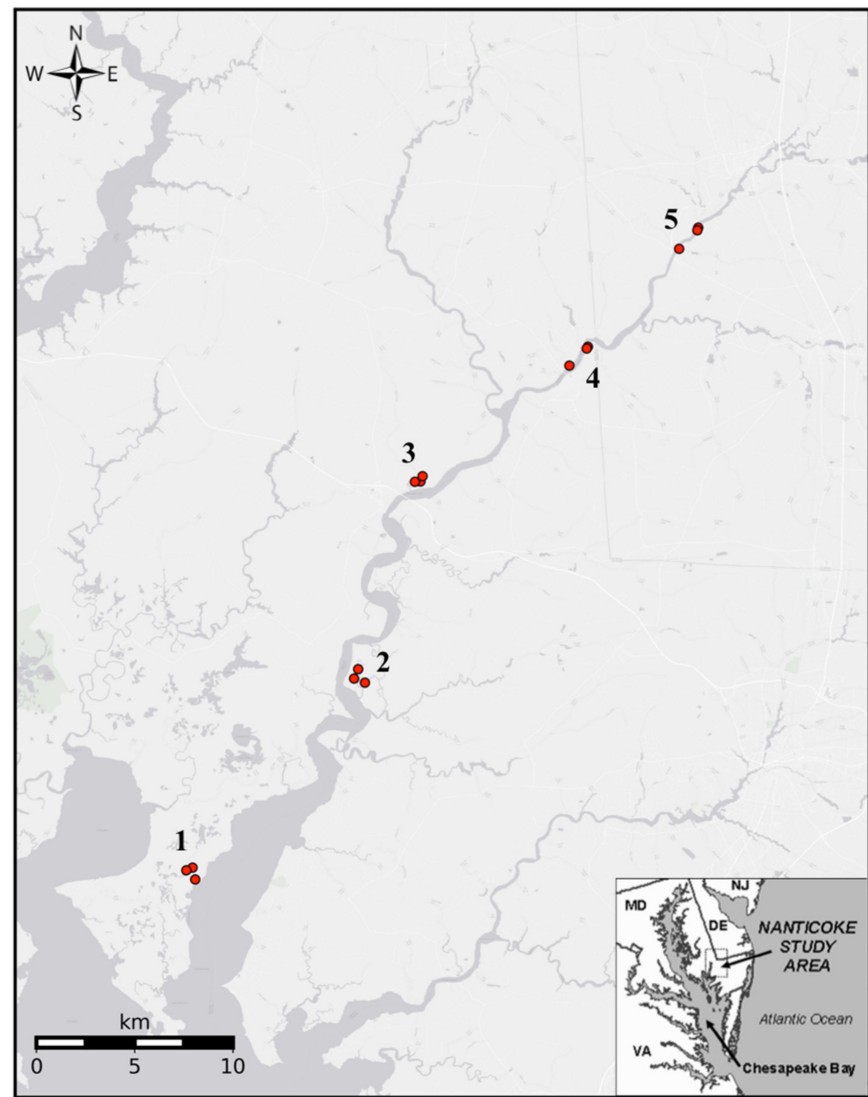

**Figure 1.** Location of study sites along the Nanticoke River. Each site contains three subsites.

**Table 1.** Salinity regime, average salinity, distance from the mouth along the river channel, and the dominant vegetation for each site along the Nanticoke River (average salinity data from [30]; plant nomenclature is according to the United States Deparment of Agriculture PLANTS database, https://plants.usda.gov, accessed on 28 May 2021).

| Site | Salinity Regime | Average Salinity (ppt) | Distance from Mouth (km) | Dominant Vegetation |
|:---:|:---:|:---:|:---:|:---:|
| 1 | Mesohaline | 10 | 12 | *Juncus roemerianus, Spartina alterniflora, Distichlis spicata, Spartina patens, Bolboschoenus robustus* |
| 2 | Mesohaline | 5 | 25 | *Spartina alterniflora, Spartina patens, Spartina cynosuroides, Distichlis spicata, Bolboschoenus robustus* |
| 3 | Oligohaline | 2 | 40 | *Peltandra virginica, Mikania scandens, Bidens laevis, Typha* sp., *Polygonum punctatum* |
| 4 | Oligohaline | 1 | 50 | *Polygonum arifolium, Impatiens capensis, Bolboschoenus fluviatilis, Peltandra virginica, Pilea pumila* |
| 5 | Tidal Freshwater | 0.1 | 60 | *Peltandra virginica, Polygonum arifolium, Zizania aquatica, Impatiens capensis, Nuphar lutea* |

Sediment cores were collected from each of the 15 subsites in August 2008. Cores were taken from a random location approximately 10 m from the SETs using a McAuley corer, which collects a 100 cm by 7.5 cm diameter half-cylinder core while minimizing vertical compaction. Each core was divided into 2.5 cm sections within the top 10 cm, 5 cm sections from 10 to 50 cm, and 10 cm sections from 50 to 100 cm. Samples were brought back to the laboratory, dried at 70° C for 48 h and analyzed for water content and bulk density. The dried soil was then placed in a muffle furnace at 400° C for 16 h to determine organic matter content by loss on ignition (LOI) [33].

Average bulk density and organic matter for the entire 100 cm core was calculated using the following steps: (1) averaging the upper four 2.5 cm sections to get an average for the top 10 cm; (2) averaging the two 5 cm sections together for each of the four 10 cm interments from 10 to 50 cm; and (3) averaging these five 10 cm increments with the five 10 cm increments from the 50 to 100 cm depth.

Sediment cores were dated using the radioisotope lead-210 ($^{210}$Pb, $T_{1/2}$ = 22.3 yr), a naturally occurring isotope deposited from the atmosphere [34]. Dried ground sediment was digested with nitric and hydrochloric acids for the analysis of $^{210}$Po, a $^{210}$Pb daughter nuclide [35], with $^{209}$Po used as a yield tracer.

Accretion rates were calculated using the Constant Initial Concentration (CIC) model [36] which assumes constant inputs of $^{210}$Pb and sedimentation rates through time. Excess (unsupported) $^{210}$Pb activity will decrease exponentially with depth in the sediment [12]. The unsupported $^{210}$Pb was calculated by subtracting the supported $^{210}$Pb activity estimated by the asymptote. A linear regression of the natural log of unsupported $^{210}$Pb versus cumulative mass (g cm$^{-2}$) was used to calculate sedimentation rates in terms of annual mass burial (g cm$^{-2}$ yr$^{-1}$). Marsh accretion rates were determined by dividing the sedimentation rate by the average bulk density (g cm$^{-3}$), generating depth-based rates (cm y$^{-1}$).

Total mass accumulation rates were calculated using the sedimentation rates (g cm$^{-2}$ yr$^{-1}$) determined from the $^{210}$Pb profiles. Organic accumulation rates (g cm$^{-2}$ yr$^{-1}$) for each site were determined by multiplying the sedimentation rate by the average organic content (determined as LOI, averaged over a depth of 100 cm). Similarly, inorganic accumulation (g cm$^{-2}$ yr$^{-1}$) was calculated by multiplying the sedimentation rate by the inorganic content averaged over depth of 100 cm.

The calculation of total marsh accretion attributable to organic and inorganic matter followed Bricker-Urso et al. [12]:

$$S_i = (S_t)(LOI)/D_i \qquad (1)$$

where $S_t$ = total average sediment accumulation (g cm$^{-2}$ yr$^{-1}$), LOI = ratio of loss on ignition (%LOI/100 for organic, 1−%LOI/100 for inorganic), $D_i$ = sediment density (1.1 g cm$^{-3}$ for organic and 2.6 g cm$^{-3}$ for inorganic) [37], and $S_i$ = organic or inorganic sediment accretion (cm yr$^{-1}$). The amount of accretion due to water/porespace was also determined by subtracting both the organic and inorganic accretion from the total accretion.

Analysis of variance (ANOVA) was used to test for differences between sites in bulk density, organic matter, organic and inorganic accumulation, and accretion rates. Tukey's Honestly Significant Difference test was used to make post-ANOVA comparisons. (SAS 9.2, SAS Institute Inc., Cary, NC, USA). Linear regressions were used to explore the relationship between accretion vs. bulk density, accretion vs. organic matter, and accretion vs. distance from the mouth of the river. Additionally, the relationships between accretion vs. organic accumulation, accretion vs. inorganic accumulation, organic vs. inorganic accumulation, and percent water vs. the log of percent organic matter were tested. A curvilinear relationship was tested for organic matter vs. bulk density (R stats package, version 3.5.1, R Core Team 2018, R Foundation for Statistical Computing, Vienna, Austria). Unless otherwise noted, $p \leq 0.05$ was used as the critical value for all tests of significance.

## 3. Results

### 3.1. Soil Characteristics

Bulk density and organic matter content did not differ between the sites ($F_{4, 10} = 1.99$, $p = 0.17$; $F_{4, 10} = 0.61$, $p = 0.67$, respectively). Additionally, there was no linear relationship between accretion rate and bulk density or organic matter content ($p = 0.56$, $p = 0.96$, respectively). The soil profiles follow a general pattern of increasing bulk density and decreasing organic matter with depth, with high variability between subsites (Figure 2). Across the individual subsites, bulk density values ranged from $0.1170-0.3753$ g cm$^{-3}$ and organic matter content ranged from 21%-57% (Table 2). Overall, there was an inverse relationship between bulk density and organic matter ($p < 0.001$, R$^2$ = 0.52, Figure 3a).

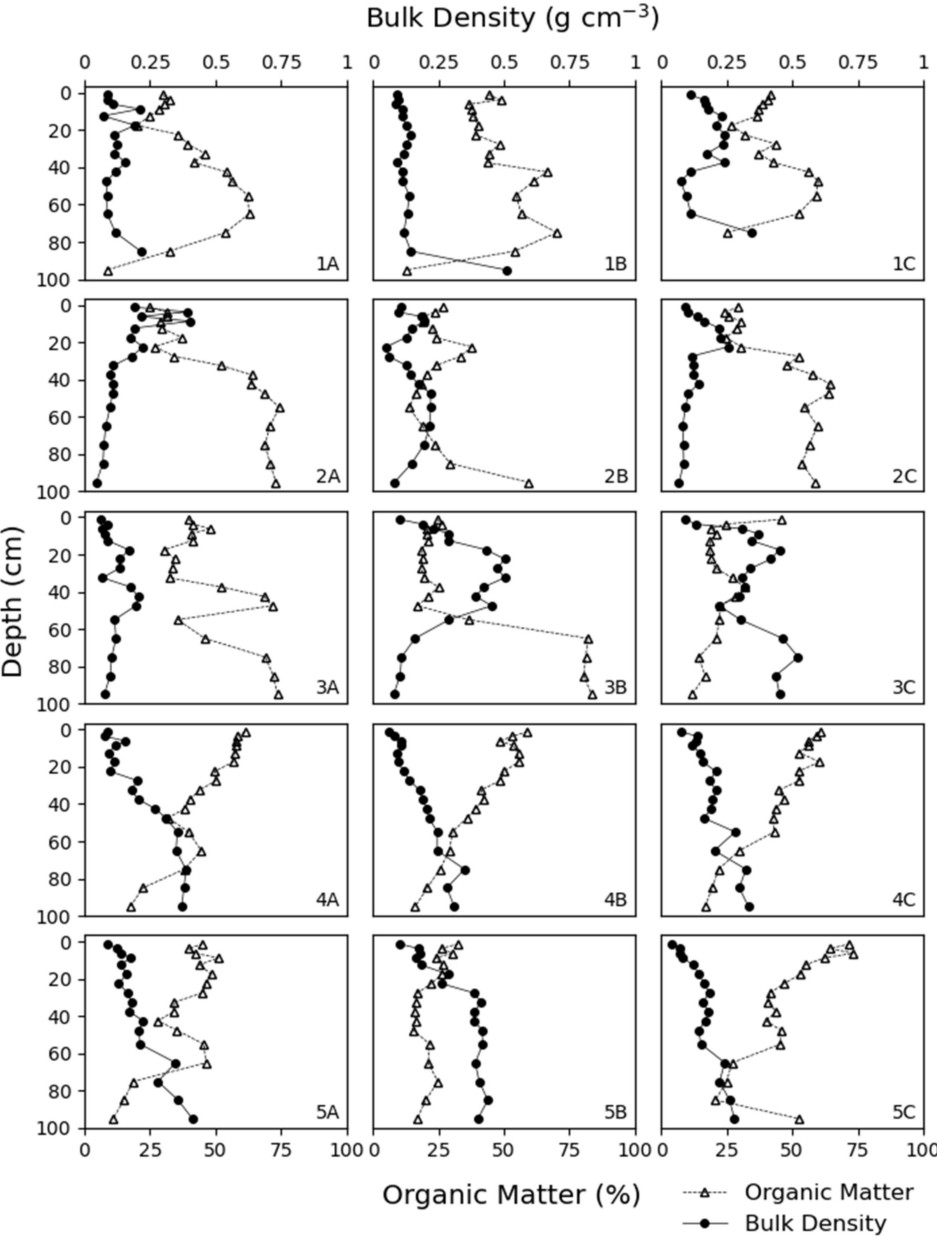

**Figure 2.** Bulk density (g cm$^{-3}$) and organic matter (%) values vs. depth across 100 cm cores for sites 1-5 (subsites A, B, C) along the Nanticoke River.

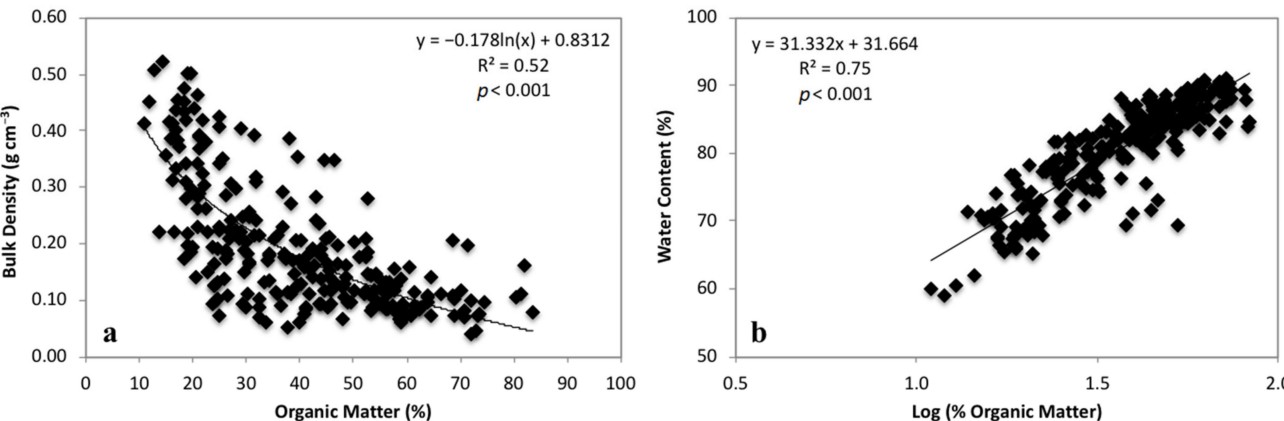

**Figure 3.** Relationships between (**a**) organic matter (%) and bulk density (g cm$^{-3}$); and (**b**) the log of percent organic matter versus percent water for soil cores taken at each subsite along the Nanticoke River.

### 3.2. Accretion Rates

The 15 soil cores had $^{210}$Pb profiles suited for measurement of sedimentation (Figure 4). The linear regressions of the ln excess $^{210}$Pb vs. cumulative mass yielded $R^2$ values ranging from 0.381 to 0.965 ($p < 0.05$ except for cores 3C, 5A, and 5B, Figure S1).

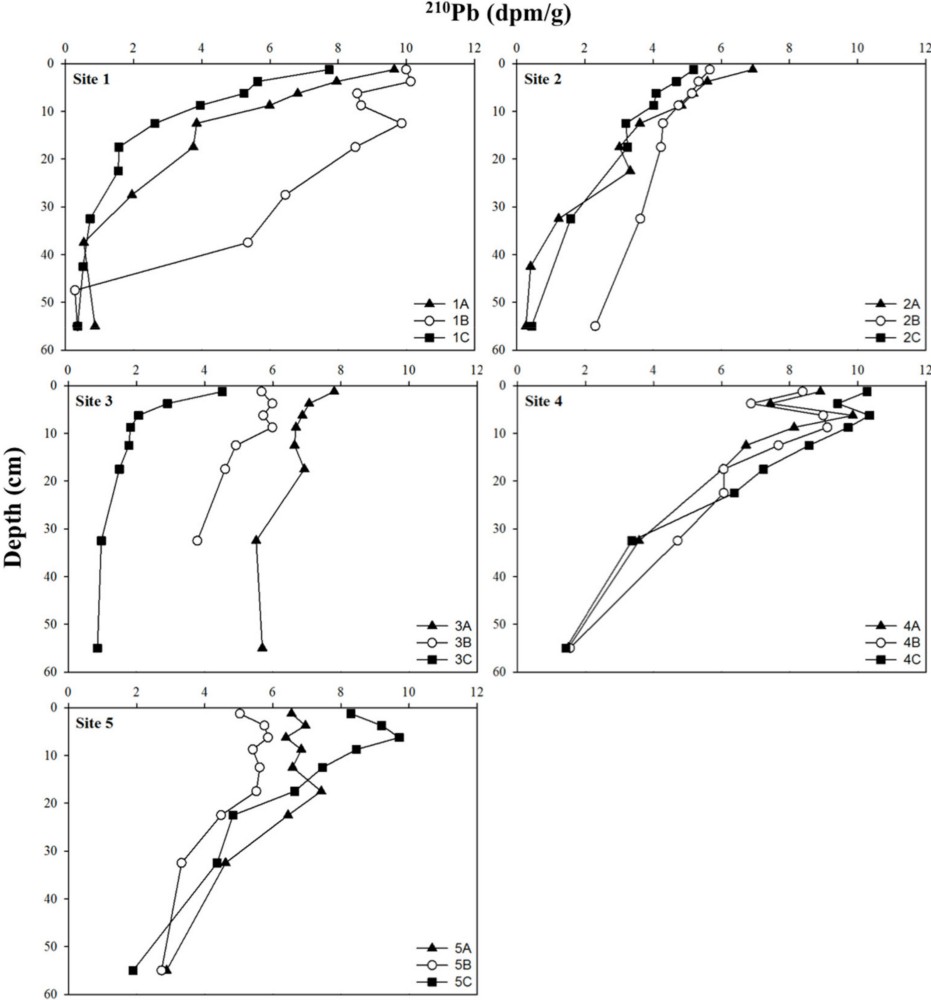

**Figure 4.** Lead-210 profiles for each site along the estuarine gradient of the Nanticoke River.

Mean accretion rates for the sites ranged from 0.240 to 0.626 cm yr$^{-1}$ (Figure 5) and with no difference in accretion rates between sites ($F_{4, 10} = 1.74$, $p = 0.22$). The accretion rates for the individual subsites were highly variable. At sites 1 and 2 (mesohaline marshes), two out of the three subsites had rates of accretion higher than RSLR (0.377 cm yr$^{-1}$ for Cambridge, MD, NOAA water level station 8571892), while at site 5 (tidal freshwater marshes), all of the subsites had accretion rates higher than RSLR. In contrast, site 3 oligohaline subsites had accretion rates lower than RSLR, and at site 4 (oligohaline marshes), two out of the three sites had accretion rates below RSLR. Despite differences in the rates of accretion across the different estuarine positions, there was no relationship between accretion and distance from the mouth of the river ($p = 0.76$). There was also no relationship between accretion and bulk density or organic matter concentrations ($p = 0.56$, 0.96, respectively).

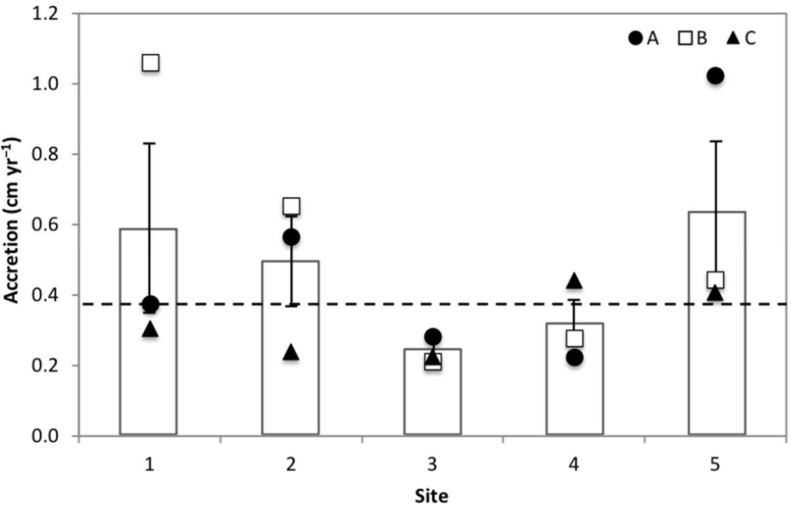

**Figure 5.** Accretion rates (cm yr$^{-1}$) for each site along the Nanticoke River (mean ± SE). Symbols indicate accretion rates for each subsite. Dotted line indicates RSLR for Cambridge, Maryland (0.377 cm yr$^{-1}$) from 1943 to 2018 obtained from http://www.tidesandcurrents.noaa.gov/ accessed on 13 May 2019.

**Table 2.** Vertical accretion, bulk density and organic matter (averaged over 100 cm core ($n = 10$)), accumulation rates, and the percent of accretion that is contributed by organic (Org) and inorganic matter (Inorg) and water/porespace (W/P) for each subsite along the Nanticoke River.

| Sub-Site | Accretion (cm/yr) | Bulk Density (g cm$^{-3}$) | Organic Matter (%) | Accumulation Rates (g cm$^{-2}$ y$^{-1}$) | | Percent of Accretion | | |
|---|---|---|---|---|---|---|---|---|
| | | | | Org | Inorg | Org | Inorg | W/P |
| 1A | 0.3772 | 0.1242 | 44.49 | 0.0210 | 0.0262 | 5.07 | 2.68 | 92.26 |
| 1B | 1.0611 | 0.1618 | 48.11 | 0.0745 | 0.0803 | 6.38 | 2.91 | 90.71 |
| 1C | 0.3045 | 0.1835 | 42.95 | 0.0221 | 0.0293 | 6.59 | 3.70 | 89.71 |
| 2A | 0.5674 | 0.1265 | 57.49 | 0.0686 | 0.0507 | 11.00 | 3.44 | 85.57 |
| 2B | 0.6537 | 0.1539 | 26.59 | 0.0253 | 0.0699 | 3.52 | 4.12 | 92.36 |
| 2C | 0.2397 | 0.1195 | 49.62 | 0.0172 | 0.0175 | 6.53 | 2.81 | 90.66 |
| 3A | 0.2833 | 0.1170 | 52.07 | 0.0134 | 0.0124 | 4.31 | 1.68 | 94.01 |
| 3B | 0.2133 | 0.2683 | 46.67 | 0.0289 | 0.0330 | 12.32 | 5.96 | 81.72 |
| 3C | 0.2240 | 0.3753 | 20.68 | 0.0134 | 0.0512 | 5.42 | 8.80 | 85.78 |
| 4A | 0.2259 | 0.2687 | 40.51 | 0.0135 | 0.0198 | 5.44 | 3.38 | 91.18 |
| 4B | 0.2781 | 0.2142 | 35.99 | 0.0121 | 0.0215 | 3.95 | 2.97 | 93.08 |
| 4C | 0.4415 | 0.2285 | 38.73 | 0.0279 | 0.0441 | 5.74 | 3.84 | 90.43 |
| 5A | 1.0252 | 0.2431 | 33.86 | 0.0521 | 0.1019 | 4.62 | 3.82 | 91.56 |
| 5B | 0.4445 | 0.3571 | 21.10 | 0.0228 | 0.0852 | 4.66 | 7.37 | 87.97 |
| 5C | 0.4081 | 0.1857 | 42.26 | 0.0193 | 0.0264 | 4.31 | 2.49 | 93.20 |

### 3.3. Mass Accumulation Rates

Organic and inorganic accumulation rates did not differ between sites ($F_{4, 10} = 0.71$, $p = 0.61$; $F_{4, 10} = 1.12$, $p = 0.40$ respectively), but were positively related to each other ($p < 0.05$, $R^2 = 0.37$, Figure 6a).

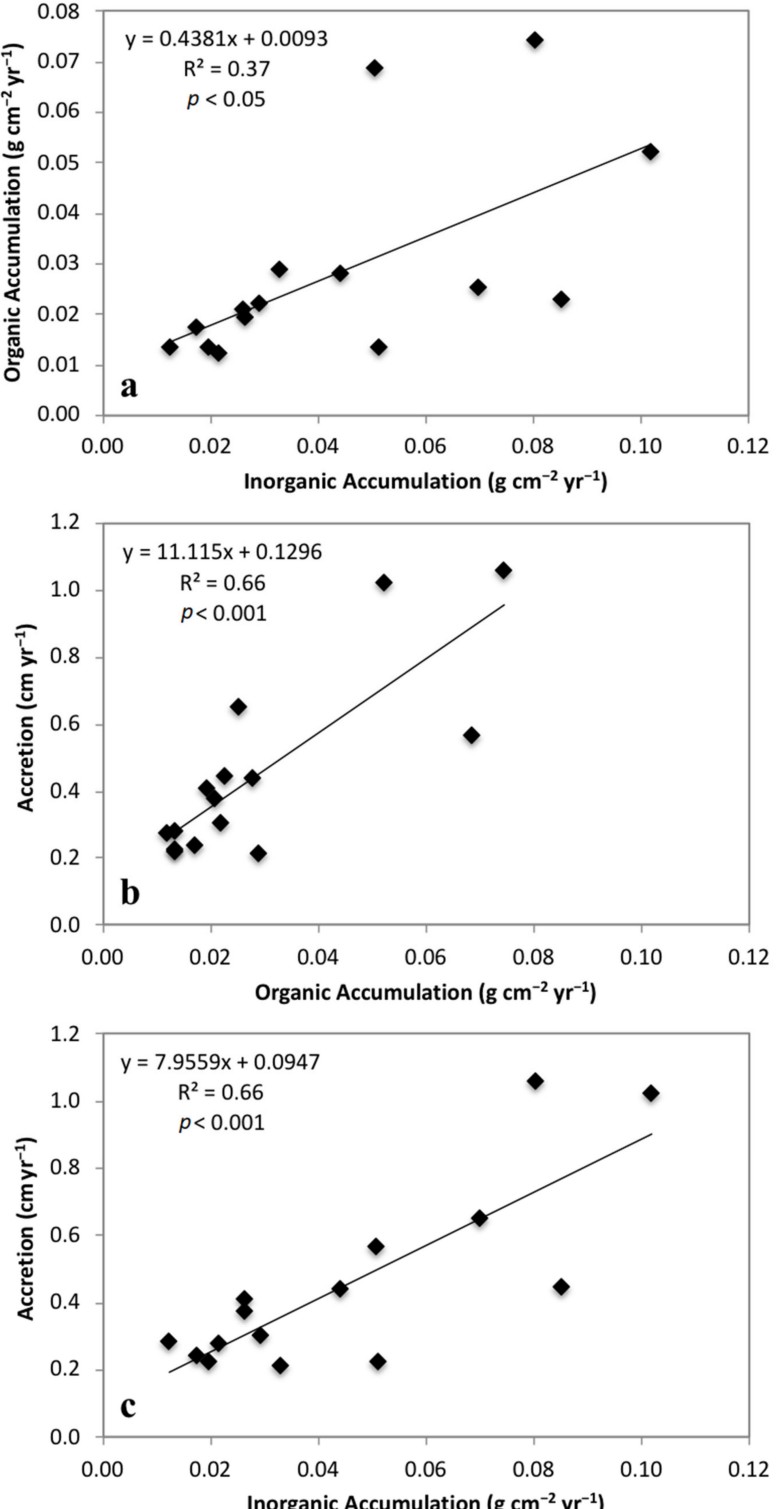

**Figure 6.** Relationships between (**a**) organic accumulation (g cm$^{-2}$ yr$^{-1}$) and inorganic accumulation (g cm$^{-2}$ yr$^{-1}$); (**b**) accretion (cm yr$^{-1}$) and organic accumulation (g cm$^{-2}$ yr$^{-1}$); and (**c**) accretion (cm yr$^{-1}$) and inorganic accumulation (g cm$^{-2}$ yr$^{-1}$) for subsites along the Nanticoke River.

Overall, most subsites received similar contributions of organic and inorganic accumulation (Table 2). Subsites 2B, 3C, 4B, 5A, and 5B had 2−3 times more inorganic accumulation than organic; however, when looking at the amount each component contributes to vertical accretion, only three of these sites (2B, 3C, and 5B) had a higher percentage of inorganic than organic contribution. At the remaining 12 sites, organic matter contributed approximately 1.5−3 times more towards accretion than did the inorganic component. Water/porespace contributed the largest percentage towards accretion, accounting for 82%−94% of the total accretion (Table 2).

A positive relationship between both organic and inorganic accumulation rates and accretion rates was observed ($p < 0.001$, Figure 6b,c). Organic and inorganic accumulation each explained about 66% of the variability in vertical accretion rates. The slope coefficient for accretion vs. organic accumulation was higher than the slope coefficient for accretion vs. inorganic accumulation, suggesting that the input of organic matter resulted in a greater increase in volume than the same input of mineral matter per unit mass.

## 4. Discussion

Accretion rates were expected to be highest in tidal freshwater marshes and decrease downstream as salinity levels increased; however, accretion rates were highly variable across the Nanticoke River subestuary. Only eight out of the 15 subsites had accretion rates higher than RSLR for the area (0.377 cm yr$^{-1}$ for Cambridge, Maryland). The highest rates of accretion were found in the tidal freshwater and mesohaline sites (sites 1, 2, and 5). The oligohaline marshes found in the mid-estuary sites (sites 3, 4) had the lowest rates of accretion (Figure 5), with only one of the subsites (4C) accreting at a rate faster than RSLR for the area. A previous study done on the Nanticoke River using surface elevation tables (SET) to measure elevation changes also found that the mid-estuarine sites are losing elevation despite high rates of accretion [30]. Noe et al. [38] also found that long-term accretion rates measured using $^{210}$Pb dating declined downstream from the tidal freshwater forested wetlands to the oligohaline marshes, however that study only examined sites in the upper estuary. The high accretion rates found in the tidal freshwater marshes at site 5 are likely due to riverine sediment input and are consistent with accretion rates found by Kearney and Ward [23] using pollen and $^{210}$Pb analyses. They found that tidal freshwater marshes had the highest rates of accretion in the estuary, more than doubling since European settlement of the watershed.

The decreasing rates of accretion with increasing salinity in estuaries of varying tidal amplitude along the Atlantic coast of the U.S. [23–25], were not observed along the Nanticoke River. As sediment gets trapped in the in the upper reaches of the estuary, accretion rates generally decline down estuary [23]. As a result, the oligohaline marshes located mid-estuary receive little allochthonous sediment input [29]. The low accretion rates found at both sites 3 and 4 on the Nanticoke River suggest that these marshes are receiving little sediment input. Further, the marshes at site 3 are located in tidal creeks off of the main channel of the river, so these marshes are likely accreting more slowly due to less frequent tidal flooding [26,39].

The Nanticoke River is ebb dominated and microtidal, and river dynamics may overwhelm weak tidal currents and increase net sediment transport [26,27], so the marshes in the lower estuary are more likely dependent on autochthonous organic matter production or sediment brought in from major storm events. Therefore, it is surprising that in the lower reaches of the estuary, the brackish marshes had accretion rates similar to the tidal freshwater marshes. The lack of a clear trend across the estuary and the variability of accretion rates across the subsites indicates that the mechanisms controlling accretion dynamics are complex and likely driven by site-specific factors rather than estuary-wide factors.

The contribution of organic and inorganic matter to accretion can vary depending on the location of the tidal marsh within the estuary. Vertical accretion in salt marshes is largely driven by organic matter accumulation [17,20], while in tidal freshwater marshes,

both mineral and organic matter accumulation influence accretion [11]. It was expected that inorganic accumulation would have a larger influence on accretion in the tidal freshwater marshes while organic accumulation would play a larger role downstream in the brackish marshes. Instead, most sites had similar rates of organic and inorganic accumulation, with both explaining 66% of the variability in marsh accretion rates (Figure 6b,c). However, when looking at the percent each component contributes to vertical accretion, the organic contribution was higher at 12 out of the 15 sites (Table 2). Further, the water/porespace accounted for 82%−94% of the total accretion (Table 2), and the relationship between loss-on-ignition and water content in the marsh cores suggests that the water retained by the marsh sediment is associated with the organic component (Figure 3b). When associated waters were considered, organic matter accounted for 91%−98% of vertical accretion across the estuary (Figure 7) [12]. Additionally, the slope coefficient for accretion vs. organic accumulation was higher than the slope coefficient for accretion vs. inorganic accumulation, resulting in a greater increase in volume than the same input of mineral matter (Figure 6b,c) [11]. These findings suggest that organic matter accumulation has a large influence on accretion across the estuary regardless of location of the marsh.

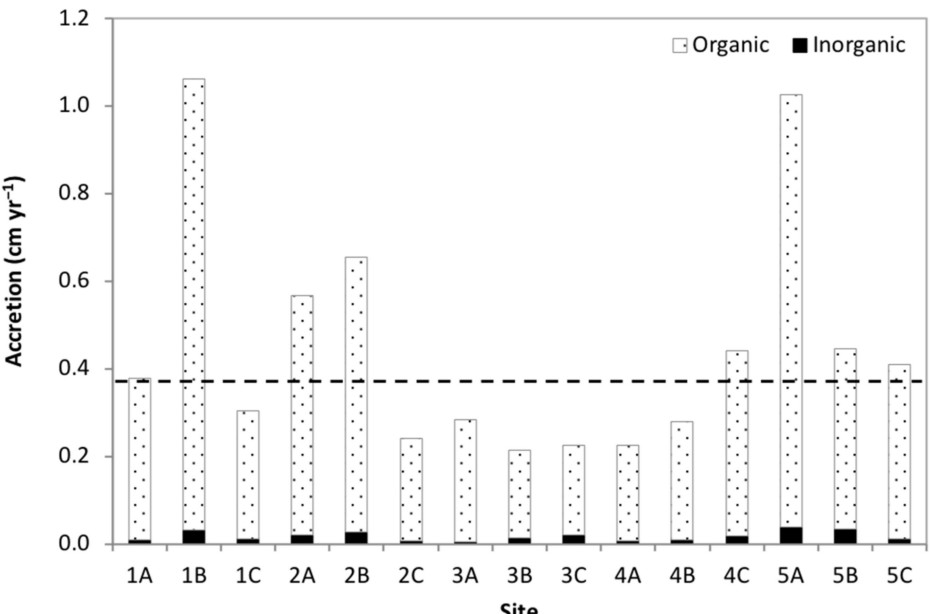

**Figure 7.** Vertical accretion rates (cm yr$^{-1}$) along the Nanticoke River with contributions from organic (including water/porespace) and inorganic inputs. Dotted line indicates RLSR for Cambridge, Maryland (0.377 cm yr$^{-1}$) from 1943 to 2018 obtained from http://www.tidesandcurrents.noaa.gov/, accessed on 13 May 2019.

Although we found that organic matter accumulation plays a large role in vertical accretion, inorganic sediment may also play an important role [13]. The biogeomorphic feedbacks between inorganic sediment and the plant biomass strongly influence the ability of a marsh to keep pace with sea level rise [40]. The presence of plants on the marsh surface enhances inorganic sedimentation [41], and increased nutrient-rich sediment enhances plant growth [9,42]. However, the elevation of the marsh also plays a role, and plants have an optimum elevation at which they are most productive [9]. If the marsh elevation is lower than the optimum elevation for plant growth, an increase in the depth of tidal flooding will lead to a decrease in plant productivity, and thus a decrease in sedimentation [40]. The positive relationship between organic and inorganic accumulation across the sites supports the idea that both inputs are important for maintaining marsh surface elevation (Figure 6a). In order to avert marsh submergence, accretion deficits that cannot be offset by increased organic accumulation must be accompanied by increased sediment accumula-

tion [22]. Along the Nanticoke River, inorganic inputs may be too low and organic matter accumulation may not be enough to keep these marshes above relative sea level.

## 5. Conclusions

Nanticoke River tidal marshes do not fit the broad paradigm of higher accretion rates at lower salinities. Accretion rates were highly variable across the estuarine gradient with the lowest rates of accretion found in the oligohaline marshes. Only 8 out of the 15 subsites across the Nanticoke River subestuary had accretion rates higher than RSLR for the area. Organic matter accumulation had a large influence on accretion rates across the estuary, regardless of the location. Interestingly, inorganic sedimentation was also important across the estuary, suggesting that both inputs play a key role in marsh surface elevation in these marshes. Organic matter contribution is especially important in microtidal marshes with low mineral sediment supply, particularly mid-estuarine oligohaline marshes. If organic matter accumulation is not sufficient, these marshes are at risk of being lost to rising sea levels. The loss of these tidal marshes could result in the loss of important ecosystem services such as storm surge and flood attenuation, water purification, and carbon sequestration [43].

**Supplementary Materials:** The following are available online at https://www.mdpi.com/article/10.3390/jmse9070751/s1, Figure S1: Linear regression of the natural log of unsupported $^{210}$Pb versus cumulative mass (g cm$^{-2}$) for each subsite along the Nanticoke River.

**Author Contributions:** Conceptualization, J.R.A. and A.H.B.; methodology, J.R.A., A.H.B. and J.C.C.; field sampling, J.R.A. and A.H.B.; sedimentary analyses, J.C.C. and J.R.A.; formal analysis, J.R.A., A.H.B., and J.C.C.; writing—original draft preparation, J.R.A.; writing—review and editing, J.R.A., A.H.B. and J.C.C.; visualization, J.R.A.; project administration, A.H.B.; funding acquisition, A.H.B., J.C.C., and J.R.A. All authors have read and agreed to the published version of the manuscript.

**Funding:** This research was funded by the Department of Energy National Institute for Climatic Change Research (NICCR).

**Institutional Review Board Statement:** Not applicable.

**Informed Consent Statement:** Not applicable.

**Data Availability Statement:** The datasets generated and analyzed during this study are available from the corresponding author upon request.

**Acknowledgments:** We thank Mike Owens for assistance with the Lead-210 dating. We also thank Scott Allen for assistance with sample analysis in the laboratory.

**Conflicts of Interest:** The authors declare no conflict of interest. The funders had no role in the design of the study; in the collection, analyses, or interpretation of data; in the writing of the manuscript, or in the decision to publish the results.

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
