# Peer review of "Contributions of Organic and Mineral Matter to Vertical Accretion in Tidal Wetlands across a Chesapeake Bay Subestuary"

_jmse, doi:10.3390/jmse9070751_

Round 1
Reviewer 1 Report
I was pleased to read the manuscript. The presented study covers a current environmental topic. It delivers exciting results on the vertical accretion of wetlands in the Chesapeake Bay and the survival of coastal and estuarine wetlands in the face of rising ocean water level. The study is based on a well-prepared experiment; the results are well documented, adequately elaborated, described, and discussed. In the discussion section, I miss some points regarding possible forecasts of changes in tidal wetlands - loss in the area, biodiversity, maybe some remark about the consequences of climate changes for this type of wetlands in the US and elsewhere? My comments do not detract from the overall positive assessment of the manuscript that in my opinion may be published in its present form.
Author Response
Point 1: In the discussion section, I miss some points regarding possible forecasts of changes in tidal wetlands -loss in the area, biodiversity, maybe some remark about the consequences of climate changes for this type of wetlands in the US and elsewhere? My comments do not detract from the overall positive assessment of the manuscript that in my opinion may be published in its present form.
Response 1: Thank you for your review. A sentence has been added at the end of the conclusion to tie in the importance of tidal wetlands and why losing them is a concern (lines 359-361).
Reviewer 2 Report
First Paragraph should better explain and differentiate eustatic and relative SLR. Is the 0.08-0.22 mm on top of eustatic or is it the RSLR?
Line 44-45 needs its own citation.
Please explain more about lead 210 dating in the introduction, perhaps a paragraph with papers using the method in wetlands.
This isn't on you but spacing is off on Figure 1 caption and Table 1 transition.
In Figure 2 there is a need to restate the subplots in the caption of this figure
Line 168..is SLR relative or eustatic. Be consistent throughout paper.
Any person that has dealings with SETs will notice the lack of comparison between sedimentation measured by the SETs and lead. You mention SETs in the methods section as if this was to be a complementary study to the SET study yet you do not actually present and comparing data.
The discussion should include at least one paragraph comparing lead results to other studies that have used the same methods.
Overall this paper is publishable and valuable in the current form, however there is room for improvement by directly comparing lead accretion results with SET accretion results.
Author Response
Point 1:First Paragraph should better explain and differentiate eustatic and relative SLR. Is the 0.08-0.22 mm on top of eustatic or is it the RSLR?
Response 1:To be consistent with the references cited in line 26, the term global mean sea level was used. We added in “mean” to make sure it is clear. In all other places, we are referring to RSLR and the text has been updated to be clear and consistent.
Point 2:Line 44-45 needs its own citation.
Response 2:Citations were updated.
Point 3:Please explain more about lead 210 dating in the introduction, perhaps a paragraph with papers using the method in wetlands.
Response 3:Lines 64 and 65ofintroduction were updated to include citations of studies done in wetlands using lead 210 dating.
Point 4: This isn't on you but spacing is off on Figure 1 caption and Table 1 transition.
Response 4:Thank you for catching the formatting issue. We will work with the editors to ensure the formatting is correct.
Point 5: In Figure 2 there is a need to restate the subplots in the caption of this figure
Response 5: Figure caption has been updated.
Point 6: Line 168..is SLR relative or eustatic. Be consistent throughout paper.
Response 6:Updated on line 168 and throughout the paper to be consistent.
Point 7: Any person that has dealings with SETs will notice the lack of comparison between sedimentation measured by the SETs and lead. You mention SETs in the methods section as if this was to be a complementary study to the SET study yet you do not actually present and comparing data.
Response 7:The lead measurements were not meant to be a complementary study to the SET study which is why we did not present any of the SET data. To avoid any confusion, were moved the sentence referring to the SET sites in the methods section(line 88).
Point 8: The discussion should include at least one paragraph comparing lead results to other studies that have used the same methods.
Response 8:Lines 230 and 234 were updated to clearly describe that lead 210 analyses were used in the studies referenced.
Point 9: Overall this paper is publishable and valuable in the current form, however there is room for improvement by directly comparing lead accretion results with SET accretion results.
Response 9:Thank you for your review. The focus of the paper was to examine long term accretion rates and determine the contribution of the organic and inorganic matter to accretion. We were not intending to directly compare our results to the SET accretion results and based on your comment above regarding the mention of the SETs in the methods, we removed the sentence from the methods to avoid any confusion.